# The Impact of Antiseptic-Loaded Bacterial Nanocellulose on Different Biofilms—An Effective Treatment for Chronic Wounds?

**DOI:** 10.3390/jcm11226634

**Published:** 2022-11-09

**Authors:** Hanna Luze, Ives Bernardelli de Mattos, Sebastian Philipp Nischwitz, Martin Funk, Alexandru Cristian Tuca, Lars-Peter Kamolz

**Affiliations:** 1Division of Plastic, Aesthetic and Reconstructive Surgery, Department of Surgery, Medical University of Graz, Auenbruggerplatz 29/2, 8036 Graz, Austria; 2Fraunhofer Institute for Silicate Research ISC, Translational Center Regenerative Therapies, 97070 Würzburg, Germany; 3EVOMEDIS GmbH, 8036 Graz, Austria; 4COREMED—Cooperative Centre for Regenerative Medicine, Joanneum Research Forschungsgesellschaft mbH, 8010 Graz, Austria; 5Research Unit for Safety in Health c/o Division of Plastic, Aesthetic and Reconstructive Surgery, Department of Surgery, Medical University of Graz, 8036 Graz, Austria

**Keywords:** antimicrobial dressings, antiseptic, bacterial nanocellulose, biofilm, chronic wounds

## Abstract

**Introduction**: Pathogenic biofilms are an important factor for impaired wound healing, subsequently leading to chronic wounds. Nonsurgical treatment of chronic wound infections is limited to the use of conventional systemic antibiotics and antiseptics. Wound dressings based on bacterial nanocellulose (BNC) are considered a promising approach as an effective carrier for antiseptics. The aim of the present study was to investigate the antimicrobial activity of antiseptic-loaded BNC against in vitro biofilms. **Materials and Methods**: BNC was loaded with the commercially available antiseptics Prontosan^®^ and Octenisept^®^. The silver-based dressing Aquacel^®^Ag Extra was used as a positive control. The biofilm efficacy of the loaded BNC sheets was tested against an in vitro 24-hour biofilm of *Staphylococcus aureus* and *Candida albicans* and a 48-hour biofilm of *Pseudomonas aeruginosa*. In vivo tests using a porcine excisional wound model was used to analyze the effect of a prolonged treatment with the antiseptics on the healing process. **Results:** We observed complete eradication of *S. aureus* biofilm in BNC loaded with Octenisept^®^ and *C. albicans* biofilm for BNC loaded with Octenisept^®^ or Prontosan^®^. Treatment with unloaded BNC also resulted in a statistically significant reduction in bacterial cell density of *S. aureus* compared to untreated biofilm. No difference on the wound healing outcome was observed for the wounds treated for seven days using BNC alone in comparison to BNC combined with Prontosan^®^ or with Octenisept^®^. **Conclusions:** Based on these results, antiseptic-loaded BNC represents a promising and effective approach for the treatment of biofilms. Additionally, the prolonged exposure to the antiseptics does not affect the healing outcome. Prevention and treatment of chronic wound infections may be feasible with this novel approach and may even be superior to existing modalities.

## 1. Introduction

The biological process of successful wound healing is achieved through four precisely programmed, consecutive phases: hemostasis, inflammation, proliferation, and remodeling [1]. Chronic wounds are wounds that fail to progress through the normal consecutive phases of wound healing in an orderly and timely manner. They represent a major burden to patients, healthcare providers and healthcare systems [2]. While wounds may be colonized with a variety of microorganisms, tissue invasion or damage does not happen necessarily [3]. Characterization of this biofilm includes a variety of techniques ranging from older established methods (e.g., wound swabs, counting of bacterial colonies) to modern technologies such as fluorescent labeling and mathematical predictive modeling [4].

Shifts in the colonization flora, however, may cause pathogenic biofilms, which may be considered one of the most important factors contributing to pathological wound healing in addition to numerous potential factors like the patient’s age, nutritional status, or presence of a chronic disease or immunocompromised state [1,2]. The presence of pathogenic biofilm may also be accompanied with infection, causing local symptoms such as swelling, erythema, pain or heat [3,5].

Pathogenic biofilms usually consist of 10–20% pathogenic microbes, bacteria or fungi, that infect or invade the wound bed and 80–90% self-produced, extracellular polymeric substance [6]. Several genetic and biochemical effects within a pathogenic biofilm result in more vigorous immune responses, consequently leading to chronic inflammation compared to nonpathogenic biofilms that are effectively controlled and ultimately removed by the body’s own clearing mechanisms [7]. Pathogenic biofilms are homogeneously distributed throughout the wound bed, impeding their identification and initiation of targeted therapeutic intervention [2].

While surgical debridement is an effective option in the reduction and eradication of bacterial load [8], nonsurgical attempts to eradicate pathogenic biofilms and treat chronic wound infections are limited to the use of conventional antibiotics and antiseptics to date [6].

The use of bacterial nanocellulose (BNC) as carriers for various “active” substances might be a clinically feasible approach to address this problem. This biomaterial meets several desirable characteristics of an “ideal” wound dressing such as being non-adhesive, reducing the number of dressing changes, allowing pH modulation of the wound bed [9] or providing a moisture balance and cooling effect [10]. In this context, BNC-based wound dressings are considered a suitable carrier for different commercially available antiseptics to directly inhibit bacterial growth [11,12]. The continuous and therapeutic release of active ingredients observed in BNC-based dressings is considered beneficial for wound treatment applications due to the homeostasis of steady concentrations over the entire treatment period [11]. The present study aimed to investigate the antimicrobial activity of antiseptic-loaded BNC against in vitro biofilms of Gram-positive and -negative bacteria as well as fungi. Furthermore, an in vivo test was proposed aiming to assess the effect of a prolonged exposure of antiseptics on the wound healing outcome.

## 2. Materials and Methods

Epicite^hydro®^ 10 × 10 cm sheets (QRSkin GmbH, Würzburg, Germany; Ref-No. 800003-M02B) were used as the BNC matrix carrier for application in the in vitro biofilm models. BNC was loaded with the commercially available polyhexanide (PHMB)-based antiseptic Prontosan^®^ (B. Braun Melsungen Ag, Melsungen, Germany) or the octenidine-based antiseptic Octenisept^®^ (Schülke & Mayr GmbH, Norderstedt, Germany) as described by Bernardelli et al. [11,12]. The silver-containing dressing Aquacel^®^Ag Extra (ConvaTech Group, United Kingdom) was used as a positive control, since this dressing is known for good results against microorganisms capable of producing biofilms [13]. The effect of loaded BNC on the biofilm was tested against a 24-hour biofilm of Gram-positive bacteria (*Staphylococcus aureus*) as well as fungi (*Candida albicans*) and a 48-hour biofilm of Gram-negative bacteria (*Pseudomonas aeruginosa*). A biofilm assay that was left untreated was used as a negative control.

### 2.1. 24-Hour CDC Bioreactor Model—S. aureus

The biofilm model that was used for these experiments was created according to the adapted ASTM International Standard (E2647-13). In short, an overnight culture of *S. aureus* (ATCC 6538) was set up by inoculation of 10 mL of Tryptone Soya Broth (TSB) with a single colony of the strain. The inoculated strain was incubated at 37 °C and 125 rpm in an orbital shaking incubator. The overnight culture of *S. aureus* was then adjusted to 0.5 McFarland (~1 × 10^8^ colony-forming unit (CFU)/mL) in TSB. Then, 1 mL of the adjusted culture was used to inoculate the CDC bioreactor, which was made up to a final volume of 300 mL of TSB. The CDC bioreactor was incubated in batch phase at 37 °C and 65 rpm for 24 h. After 24 h, each epicite^hydro®^ dressing was placed in 100 mL of Prontosan^®^ and Octenisept^®^ for 30 min at room temperature. The process of loading the BNC was previously described by our group in 2019 [12]. Coupons were then briefly washed in Phosphate-Buffered Saline (PBS), then added in triplicate and incubated at 37 °C for 24 h. After 24 h of exposure, coupons were removed from the wells, added to 10 mL of Dey Engley neutralizing broth (Sigma-Aldrich, MA, USA) and sonicated for 30 min at full power. Each tube was briefly vortexed prior to sampling. Samples were added to 96-well plates, serial diluted in a ratio of 1:10 in PBS and plated onto Tryptone Soya Agar (TSA) in duplicate by pipetting 50 μL onto each half and spreading. Well plates were incubated overnight at 37 °C before enumeration of colony counts.

### 2.2. Drip-Flow Bioreactor—P. aeruginosa

Preparation of the drip-flow bioreactor was performed by adding a clean borosilicate microscope slide to each channel of the bioreactor prior to autoclavation at 121 °C for 15 min. An overnight inoculum was set up by inoculation of 10 mL of TSB with a single colony of *P. aeruginosa* (ATCC 700888) and incubation at 37 °C and 125 rpm. The overnight culture was adjusted to 0.5 McFarland (~1 × 108 CFU/mL) on the following day. Each channel of the drip-flow bioreactor was clamped to stop flow, prior to adding 15 mL 3 g/L of TSB and 1 mL of adjusted culture to each channel. The drip-flow bioreactor was incubated for 6 h in batch phase and connected to a nutrient flow of 270 mg/L TSB at 50 mL/hour per channel in the following.

Epicite^hydro®^ dressing was placed in 100 mL of Prontosan^®^ and Octenisept^®^ for 30 min at room temperature. Loaded epicite^hydro®^ dressings were then added to each of the biofilm coated microscope slides in triplicate. The drip-flow bioreactor was reconnected to the carboy and operated for another 24 h. After the challenge period, each microscope slide was scraped into 45 mL of the appropriate neutralizer in a sterile 100 mL beaker. The slide was washed with 5 mL of neutralizer to ensure the removal of the whole biofilm. Each sample was homogenized for 30 s, serial diluted 1:10 in PBS and plated in duplicate onto TSA by pipetting 50 μL onto each side of a 2-compartment plate. Well plates were incubated overnight at 37 °C before enumeration of colony counts.

### 2.3. 24-Hour CDC Bioreactor Model—C. albicans

Two overnight cultures of *C. albicans* (ATCC 10231) were set up by inoculation of 10 mL of Sabouraud Dextrose Broth (SDB) with a single colony of the strain. The inoculated strain was incubated at 37 °C and 125 rpm in an orbital shaking incubator. The overnight cultures of *C. albicans* were then centrifuged at 5600 rpm, whereby the supernatants were discarded and resuspended in 1 mL of fresh SDB to inoculate the CDC bioreactor in the following. The CDC bioreactor was made up to a final volume of 300 mL of SDB and incubated in batch phase at 30 °C and 125 rpm for 24 h. After 24 h, the rods were washed twice in PBS and individual coupons were placed into 12-well plates. Each epicite^hydro®^ dressing was placed in 100 mL of Prontosan^®^ and Octenisept^®^ for 30 min at room temperature. A 2.5 cm × 5.0 cm section of the loaded dressings was added to each coupon in triplicate, and incubated for 24 h at 30 °C. After 24 h of exposure, coupons were removed from the wells and added to 10 mL of Dey Engley neutralizing broth and sonicated at full power for 30 min. Each tube was briefly vortexed prior to sample taking. Samples were added to 96-well plates, serial diluted 1:10 in PBS and plated onto SDB in duplicate by pipetting 50 μL onto each half and spreading. Well plates were incubated for 48 h at 30 °C before enumeration of colony counts.

### 2.4. Neutralizer Effectiveness Validation Assay

A neutralizer effectiveness validation assay was conducted using a broad-spectrum neutralizer (+saponin for Prontosan^®^ WIF: EN 1276—2009) and Dey Engley neutralizing broth. An overnight culture of *P. aeruginosa* (ATCC 700888) was set up as described in Section 2.2.

The following day, Octenisept^®^ was mixed at a ratio of 1:1 with the broad-spectrum neutralizer and Dey Engley neutralizing broth while Prontosan^®^ was mixed at a ratio of 1:1 with the broad-spectrum neutralizer and saponin. A neutralizer toxicity control and a growth control of TSB only were included as well. After incubation for 10 min, the overnight culture of *P. aeruginosa* was adjusted to 0.5 McFarland (~1 × 10^8^ CFU/mL) and added to the solutions at a final concentration of 1 × 10^6^ CFU/mL prior to another incubation for 30 min. Solutions were then briefly vortexed, serial diluted 1:10 in PBS and plated out in duplicate, by pipetting 50 μL onto each half of a TSA plate and spreading. TSA well plates were incubated overnight at 37 °C before enumeration of colony counts.

### 2.5. In Vivo Tests

To assess if the healing process is affected by a prolonged treatment using octenidine-based and PHMB-based antiseptics, a porcine excisional wound model was used. The in vivo procedure was approved by The Animal Care and Use Committee (Veterinary University Vienna, Austrian Ministry of Science and Research). All the animals were treated in accordance with the recommendations of GV SOLAS (Gesellschaft für Versuchstierkunde/Society of Laboratory Animal Science, Germany). Using an electric dermatome, and under anesthesia and analgesia, 3 × 3-cm-sized wounds were generated in the dorsal part of two 3 month-old female domestic pigs (*Sus domesticus*; hybrid from Deutsche Landrasse and Deutsches Edelschwein) weighing 34 kg at the start of the experiments. The dermatome was set to excise 1.2 mm depth wounds, in order to mimic deep-partial burn wounds. To treat the wounds, 10 × 10 cm epicite^hydro®^ dressings were placed in kidney basins containing 5 times the BNC water content volume (100 mL per dressing) of Octenisept^®^ and Prontosan^®^ [11,12]. The dressings were then incubated for 30 min and 5 × 5 cm pieces were placed over the wound bed, to cover the wound completely. A total of 32 wounds, divided in the two animals, were treated using this approach. Twelve wounds were treated using epicite^hydro^ incubated with Octenisept^®^, and 12 were treated with the combination with Prontosan^®^. As a control, 8 wounds, 4 in each animal, were treated solely using the epicite^hydro^ dressings. The animals were treated for 7 days and euthanized under deep anesthesia. Then, 5 × 5 cm of the tissue comprising the wounds were extracted. The tissues were excised above the superficial fascia layer and placed in formaldehyde. During the entire procedure, photo documentation for macroscopical analysis was performed. The samples were sent to TPL path Labs GmbH (Germany), where slides were mounted, and samples were stained with Hematoxylin and Eosin (H&E). From each wound, two slides were prepared and analyzed blindly by a histopathologist. Several morphometric wound healing parameters were analyzed using the software Zen 3.3 blue edition (Zeiss Microscopy, Jena, Germany), including area, average thickness, and percentage of re-epithelialization for the regenerated epidermis and average thickness for the regenerated dermis.

### 2.6. Statistical Analysis

Raw data counts were put into Microsoft Excel and the CFU/mL was calculated. One-way ANOVA was performed as the means of inferential statistics using GraphPad Prism 7 software (GraphPad Software, Inc., San Diego, CA, USA). All statistical tests were two-tailed, and differences were considered statistically significant when *p* < 0.05.

## 3. Results

### 3.1. 24-Hour CDC Bioreactor Model—S. aureus

After a 24-h treatment of the *S. aureus* biofilms with unloaded epicite^hydro®^ dressings, a bacterial density of 2.03 × 10^5^ CFU/mL was observed. In comparison, the untreated biofilm showed a bacterial density of 1.31 × 10^7^ CFU/mL. Treatment with unloaded epicite^hydro®^ dressings therefore resulted in a statistically significant 2.0 log reduction in bacterial density (*p* = 0.0055). Biofilms treated with epicite^hydro®^ loaded with Prontosan^®^ showed a bacterial density 5.17 × 10^3^ CFU/mL, exhibiting a statistically significant 4.9 log reduction in bacterial density (*p* = 0.0055). In biofilms treated with epicite^hydro®^ loaded with Octenisept^®^ or the positive control Aquacel^®^Ag Extra, no colonies were observed, exhibiting a 7.3 log reduction in bacterial cell density (*p* = 0.0001). The bacterial density reduction of *S. aureus* with all test dressings is displayed in Figure 1.

### 3.2. Drip-Flow Bioreactor—P. aeruginosa

After a 24-hour treatment of the *P. aeruginosa* biofilms with unloaded epicite^hydro®^ dressings, a bacterial density of 1.43 × 10^9^ CFU/mL was observed. In comparison, the untreated biofilm showed a similar bacterial density of 7.47 × 10^8^ CFU/mL. Compared to the untreated biofilm, a 0.3 log increase in biofilm density was observed after the treatment with unloaded epicite^hydro®^ dressings (*p* = 0.033). Biofilms treated with epicite^hydro®^ loaded with Prontosan^®^ or Octenisept^®^ showed a bacterial cell density of 1.63 × 10^6^ CFU/mL and 1.88 × 10^6^ CFU/mL, respectively. In comparison to treatment with unloaded epicite^hydro®^ exclusively, 2.7 log (*p* = 0.0006) and 2.6 log (*p* = 0.0006) reductions in biofilm density were observed, respectively. Treatment with the positive control Aquacel^®^Ag Extra showed a bacterial density of 3.34 × 10^6^ CFU/mL (*p* = 0.0019), exhibiting a 2.3 log reduction when compared to epicite^hydro®^ (*p* = 0.003). The bacterial density reduction of *P. aeruginosa* with all test dressings is displayed in Figure 2.

### 3.3. 24-Hour CDC Bioreactor Model—C. albicans

After a 24-hour treatment of the *C. albicans* biofilms with unloaded epicite^hydro®^ dressings, a fungal density of 2.04 × 10^6^ CFU/mL was observed. In comparison, the untreated biofilm showed a fungal density of 1.45 × 10^6^ CFU/mL. Treatment with unloaded epicite^hydro®^ dressings therefore resulted in a slightly higher fungal density with a 0.1 log increase (*p* = 0.0317). In biofilms treated with epicite^hydro®^ loaded with Prontosan^®^ or Octenisept^®^ and the positive control Aquacel^®^Ag Extra, no colonies were observed, exhibiting a 6.2 log reduction in bacterial cell density (*p* = 0.0001). The fungal density reduction of *C. albicans* with all test dressings is displayed in Figure 3.

### 3.4. Effect of Prolonged Antiseptic Treatment In Vivo

To analyze whether the wound healing process would be affected by a prolonged exposure to the antiseptic solutions tested, we performed a 7-day treatment using an in vivo porcine model. During the treatment period, epicite^hydro®^ carrying the antiseptic solutions was left continuously in contact with the wound bed. An example for histological cuts prepared from the regenerated tissues at the end of the treatment can be observed in Figure 4. H&E staining offers the possibility to clearly observe the regenerated tissues, while allowing the observation and quantification of extra parameters, such as the thickness of the BNC dressing and presence of exudate at the wound bed surface. No particular histological difference was observed for the wounds treated with the BNC alone (Figure 4A) compared to the same treatment in combination with Octenisept^®^ (Figure 4B) or with Prontosan^®^ (Figure 4C). The dried aspect of the BNC dressing after 7 days (Figure 5A) indicates that its content was successfully delivered into the wound bed. All the morphometric results obtained from the analyzed tissues were comparable, with no statistical significance being observed. The wound closure rate for the wounds treated with epicite^hydro®^ and with the BNC loaded with Octenisept^®^ or Prontosan^®^ after 7 days (Figure 5B) achieved rates above 75%. Both epidermal average thickness (Figure 5C) and new dermal average thickness (Figure 5D) achieved similar results for all three treatments. The area of exudate present at the last day of treatment was also very similar for all the treatments, with no difference being observed (Figure 5E), indicating that the presence of the antiseptics did not interfere in the exudate production. Despite the BNC dressing being loaded with antiseptics comprising different formulations, in comparison to the original solution present in epicite^hydro®^, the thickness of the material at the end of the treatment was comparable (Figure 5F). This last result indicates that the evaporation process was not affected.

## 4. Discussion

Chronic wounds are typically contaminated with pathogenic biofilms causing the prolonging of the inflammatory processes of wound healing, thereby complicating the healing process [14]. The major burden of chronic wounds on patients and healthcare providers has led to extensive research on therapeutic strategies within the past decades [14]. The current standard of care for chronic wounds includes wound swabs, cleaning, dressing and, if necessary, debridement of the wound bed [15].

In addition, environment sensors, using pH, hydration, odor or optical sensors, are available to monitor and manage biofilms and detect early changes in the wound bed associated with pathological wound healing [15]. New strategies of therapeutic intervention may also include the targeting of the wound microenvironment [15].

As another, well-established approach, the dynamic concept of wound bed preparation is particularly beneficial in chronic wounds that fail to progress [16]. This concept comprises comprehensive strategies of tissue management, inflammation and infection control, moisture balance and epithelial advancement (T.I.M.E) to maximize the wound healing potential [16]. However, infection control via targeting and eradicating pathogenic biofilms is especially challenging, since biofilms can be highly tolerant and resistant to antibiotics and antiseptics [2,17].

A variety of molecular mechanisms are thought to contribute to biofilm resistance against antimicrobial agents, which may subsequently lead to treatment failure [2]. Despite the partially limited effect, recent consensus guidelines recommend topical antiseptics as a first-line therapy in the treatment of chronic wounds colonized with a pathogenic biofilm [17].

Prior studies by our research group showed the effective—and clinically feasible—loading and also the release capacity of octenidine- (Octenisept^®^) and povidone-iodine (Betaisodona^®^)-based antiseptics in BNC-based wound dressings [11,12]: the antimicrobial efficacy of antiseptic-loaded BNC against *S. aureus* was tested, whereby dose-dependency was shown [11]. In the present study, the antimicrobial activities of BNC loaded with the PHMB-based antiseptic Prontosan^®^ or the octenidine-based antiseptic Octenisept^®^ were tested against a silver-containing dressing showing excellent results in diminishing biofilms.

### 4.1. Antimicrobial Activity in Gram-Positive Bacteria

Prior in vitro studies by Zmejkosk et al. on a composite hydrogel of BNC and dehydrogenative polymer of coniferyl alcohol have already confirmed our assumptions on an antibacterial activity on *S. aureus* [18]. For a potential use with commercially available antiseptics, we were able to show an effective reduction in cell density after a 24-hour treatment of a *S. aureus* biofilm with unloaded and loaded epicite^hydro®^ dressings in the present project. Treatment with unloaded epicite^hydro®^ dressings resulted in a statistically significant reduction in bacterial density, while biofilms treated with epicite^hydro®^ loaded with Prontosan^®^ resulted in a 4.9 log reduction. Loading with Octenisept^®^ even resulted in a complete eradication of the colony, similar to the treatment with the silver-containing dressing control Aquacel^®^Ag Extra.

Other in vitro studies have already demonstrated the efficacy of PHMB-based antiseptics on Gram-positive bacteria [19]; however, a slower onset of antimicrobial activity compared to other formulations is noted [20]. Furthermore, conflicting evidence regarding the antimicrobial activity of PHMB-based antiseptics was raised [21]. These findings may explain the reduced antimicrobial effect of Prontosan^®^ compared to Octenisept^®^ or silver-based dressings. Future studies evaluating a prolonged treatment period are of utmost importance to elucidate the full potential of each antiseptic solution.

In comparison to BNC loaded with PHMB-based antiseptics, complete biofilm eradication was feasible within a 24-hour treatment with Octenisept^®^-loaded BNC or a silver-based dressing. Although silver has been positioned among first-line options for wound infection and is recommended for the treatment of pathogenic biofilms [2], bacterial resistance is well documented [22,23]. In contrast, no reports of bacterial resistance on octenidine exist to our knowledge. A recent in -vitro study by Günther et al. even demonstrated a great level of bacterial metabolic inhibition by octenidine-based antiseptics in a biofilm of methicillin-resistant *S. aureus* [24]. Based on these findings, treatment with octenidine-based antiseptics such as with Octenisept^®^ may be preferable to other options, even in biofilms of drug-resistant Gram-positive bacteria.

### 4.2. Antimicrobial Activity in Gram-Negative Bacteria

Cell density of a 48-hour *P. aeruginosa* biofilm treated with unloaded epicite^hydro®^ exclusively slightly increased after a treatment period of 24 h. However, if loaded with Prontosan^®^ or Octenisept^®^, 2.7 log and 2.6 log reductions in biofilm cell density were found compared to the unloaded epicite^hydro®^. A similar log reduction was achieved in the positive control.

Bacterial resistance against silver is rare, but it has been documented in Gram-negative bacteria including *P. aeruginosa* [25]. Contrary to this, no reports of bacterial resistance have been documented for PHMB-based antiseptics [2], which were showing the best results in the present project. A variety of studies proved the initial antibacterial effect of silver-based wound dressings on (even multi-drug resistant) *P. aeruginosa* [26,27]; however, bacterial tolerance against silver may develop over time when biofilms are evident. This mechanism may explain the initial therapeutic success also observed in our in vitro experiments. Despite a possible silver release of dressings such as Aquacel^®^Ag Extra being reported for up to two weeks [28], the potential long-term limits of silver-based dressings in chronic wounds compared to antiseptic-loaded BNC dressings are yet to be elucidated.

### 4.3. Antimicrobial Activity in Fungi

Treatment of a *C. albicans* biofilm with unloaded epicite^hydro®^ dressings resulted in a slightly higher fungal cell density. However, if loaded with Prontosan^®^ or Octenisept^®^, colonies were completely eradicated. Treatment with Aquacel^®^Ag Extra resulted in a complete eradication as well. While studies on the efficacy of PHMB-based antiseptics in *C. albicans* infections with a high organic load are lacking, significant results for octenidine-based antiseptics have recently been published by Spettel et al. [29]. The authors reported a full antimicrobial effect for *C. albicans* and other Candida strains after a contact time of only 30 s. Therefore, octenidine-based antiseptics may be preferable in fungal colonization when it comes to a routine clinical use. The combination with BNC-based dressings may be beneficial to generate a prolonged and steady concentration release of active ingredients.

Since fungi can sense suitable ambient pH for adhesion, growth and invasion, pH monitoring of a wound may be an effective approach in the prevention of colonization [30]. Especially, diabetic patients (with diabetic foot ulcers) expressing a higher pH in intertriginous areas are at risk to candidiasis, mostly with *C. albicans* [30]. After initial colonization, fungi are further increasing pH of the wound bed, resulting in a clinically manifest infection [30]. In addition to an antiseptic enhancement of BNC, pH monitoring in a sterile and easy-to-handle setting is possible [9]. When chemically functionalized with indicator dye, pH monitoring through BNC can be performed without the removal of the dressing or wound contact of a pH monitor device [9]. Potential combinations of these features in future “smart dressings” may enable a release of active substances only above a certain pH-value. Epicite^hydro®^ as an advanced wound dressing might therefore not only assist in the treatment of pathogenic biofilms, but as well in its prevention, ultimately reducing the burden of chronic wounds.

### 4.4. General Assumptions on the Use of Silver-Based Dressings vs. Antiseptic-Loaded BNC

Several differences of topical antiseptics regarding anti-biofilm efficacy, safety, and tolerability in chronic wounds have been reported [2]. The available silver-based dressings commonly used in a variety of applications due to antibacterial properties not only differ in their construction and their chemical composition of the active layer, but also in their silver forms [31]. A recent study revealed that the amount of silver contained in a dressing is often not stated by the manufacturer [31]. The authors show a potential DNA-damage of silver, released via dressings and diffused into intact porcine dermis [31]. Potential silver toxicity due to the unawareness of the amount released can therefore be detrimental to proliferating cells in already healing wounds, supporting recommendations that silver-based dressings should only be applied to highly infected, chronic wounds [31].

In comparison, BNC-based wound dressings can be used for a variety of applications (e.g., [10]) and have been shown to be nontoxic, to provide rapid tissue regeneration as well as capillary formation in the wound area [18]. As previously shown by our group, BNC can additionally be used as an effective carrier of antiseptic solutions, enabling an application in the prevention and treatment of colonized, chronic wounds [11].

### 4.5. Wound Healing Process in a Prolonged Antiseptic Treatment Period

Cell survival in vitro was already reported to be affected by various antiseptic molecules, including PHMB- and octenidine-based solutions [32]. On the other hand, reduced cytotoxicity and interesting biocompatibility indexes were observed for the same antiseptics when applied in specific concentrations [33]. The detrimental effect of the antiseptics on the viability of cells, such as fibroblasts and keratinocytes, seems to be higher in vitro when tested on isolated cells, than in a more complex system [34]. Usually, the tests are conducted exposing the cells to short periods of time (a few minutes or a couple of hours) [32,33,35,36]. To our knowledge, so far, no test has been performed for longer incubation periods nor utilizing an in vivo model to analyze how the healing processes are affected. Since the potential application against pathogens capable of producing biofilms here proposed used a 24 h treatment, we decided to analyze how the wound healing process in vivo would be affected by a continuous exposure of the antiseptics. In previous works, we were able to report a sustained delivery in vitro of the antiseptic solutions over 48 h using the BNC as a delivery platform [11,12]. In our in vivo tests, we decided to use a 7-day treatment, to guarantee that the integral BNC content would be delivered and to assess the influence of the antiseptics over a prolonged period. No significant difference was observed for all the healing parameters analyzed. Both epidermal and dermal regeneration results were comparable with the control. Wound closure rates, for instance, achieved results above 75% for all treatments, which are in line with recent reports [37]. Additionally, results for the exudate area indicate that physiological secretion was not disturbed by the presence of the antiseptics. Considering the physicochemical aspects of the BNC dressing combined with the antiseptics, the comparable results for the average thickness of the dressing at the end of the treatment suggest that the evaporation process was not impaired nor promoted. This result is important, to show that the moisture offered by the dressing remains constant even after the combination with antiseptics. An ideal moisture balance was already reported as being crucial to support the healing process [38,39,40,41]. These results confirm that the method here proposed to treat pathogens capable of producing biofilm would not negatively influence the wound healing process.

Nevertheless, eradication of pathogenic biofilms and reduction of the burden caused by chronic wounds may not be feasible via this approach exclusively. To achieve wound healing, addressing other factors regarding the patient’s general health or the wound’s physical environment is of utmost importance [42].

In addition to the delivery of antibiotics, physical treatment approaches such as hyperbaric oxygen therapy, low-level laser therapy and electrical stimulation can be used in specific cases to accelerate the healing of chronic ulcers, although inconsistent success is reported [43]. Negative pressure wound therapy is a different physical treatment approach indicated for deep chronic and very exudating wounds, based on differential suction or vacuum creation onto the wound bed to enhance fluid removal, reduce edema and alter the wound microenvironment for subsequently performing skin grafting [44]. An analysis of 11 trials by Wynn et al. showed a significant reduction of the wound surface area through negative pressure wound therapy [45], which might as well be feasible via the use of advanced dressings, e.g., polylactide wound dressings [46]. To our understanding, a holistic assessment of the patient and the wound is essential to choose the optimal strategy out of multiple to prevent or treat chronic wounds and optimally manage patients.

### 4.6. Limitations

All biofilm experiments were performed in vitro, although tested antiseptics, BNC-based and silver-based wound dressings are already available on the market. The results of the release experiments of the antiseptics may not be directly transferred into clinical practice without consideration. Since pathogenic biofilms may consist of an individual mix of bacteria and fungi, the use of agar plates is not a comparable receiving medium to an actual *clinical* wound bed. Furthermore, the interaction of exudate occurring in a (chronic) wound with BNC as well as the antiseptic solution cannot be assessed within this setting. To confirm a superior antimicrobial efficacy of the loaded BNC-based wound dressings compared to commercially available (silver-based) dressings, further clinical studies are necessary. In vivo experiments were conducted in a porcine model; however this model is already widely accepted for preclinical experiments [47]. Ultimately, there is a great demand for prevention and treatment strategies for chronic wounds, and valuable results could be retrieved from this study.

## 5. Conclusions

Available evidence suggests that the majority of chronic wounds are colonized by pathogenic biofilms hindering wound healing, resulting in ineffective treatment, burdening both the patient and the healthcare providers. BNC-based wound dressings loaded with commercially available antiseptics demonstrate a potent efficacy against biofilms formed by a variety of microbes found to be prevalent within chronic wounds, including *S aureus*, *P. aeruginosa*, and *C. albicans*. Additionally, no negative influence was observed in the wound healing parameters analyzed, showing that even a 7-day treatment using the BNC loaded with antiseptics would not disturb the normal healing process. Results are comparable to available silver-based wound dressings, whereby further advantages of BNC such as the possibility for pH-monitoring of the wound bed are to be highlighted. Antiseptic-loaded BNC-based wound dressings might therefore be an effective modality in the prevention and treatment of colonized, chronic wounds. Future studies are essential to investigate and compare the in vivo efficacy of this approach in clinical care.

## Figures and Tables

**Figure 1 jcm-11-06634-f001:**
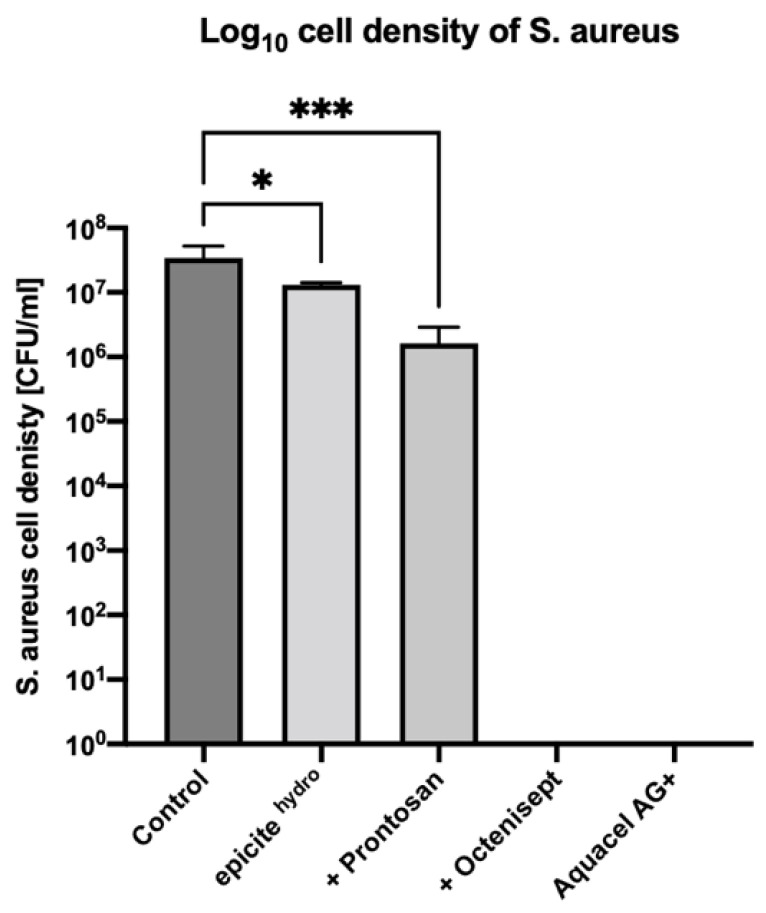
Log_10_ cell density of *S. aureus* (ATCC 6538) 24-h biofilm following 24-h treatment with the test dressings. All dressings were tested in triplicate. Error bars represent standard deviation of the mean. Asterisks represent the significant difference between the samples and the untreated control (*p* = 0.0055).

**Figure 2 jcm-11-06634-f002:**
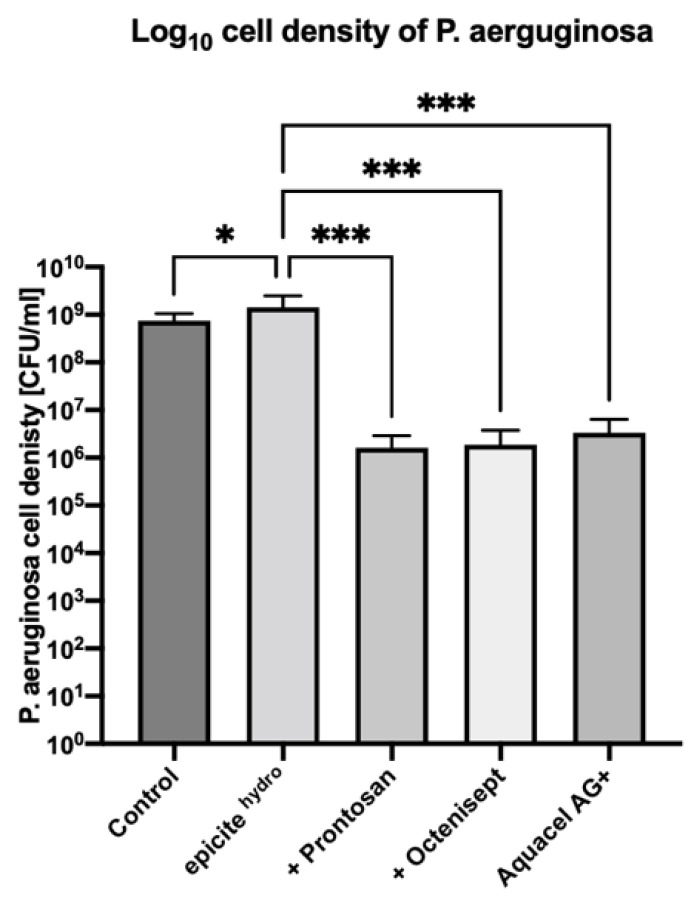
Log_10_ cell density of *P. aeruginosa* (ATCC 700888) 24-hour biofilm following 48-hour treatment with the test dressings. All dressings were tested in triplicate. Error bars represent standard deviation of the mean. Asterisks represent the significant difference between the samples and the untreated control or epicite^hydro®^ exclusively.

**Figure 3 jcm-11-06634-f003:**
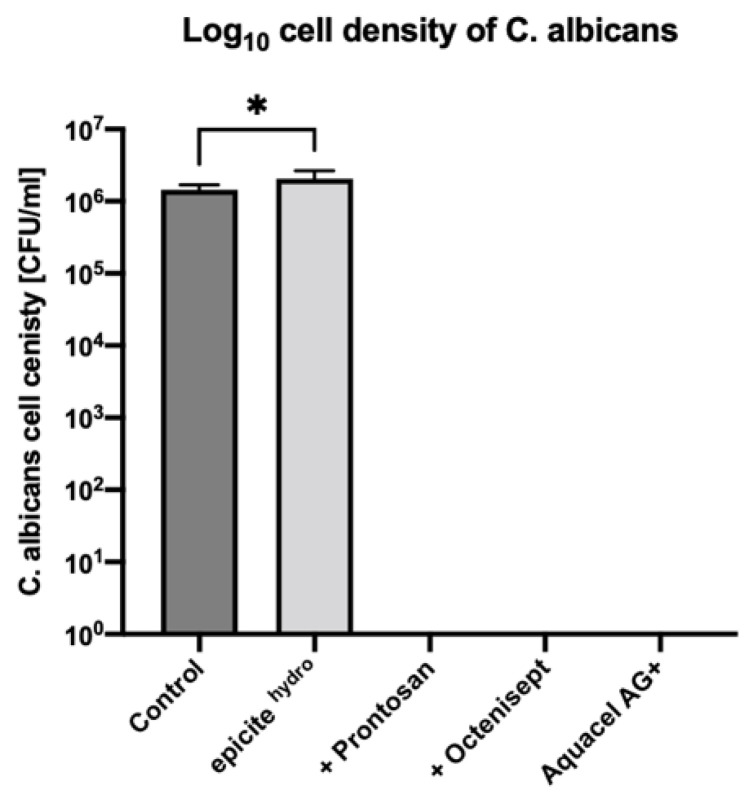
Log_10_ cell density of *C. albicans* (ATCC 10231) 24-h biofilm following 24-h treatment with the test dressings. All dressings were tested in triplicate. Error bars represent standard deviation of the mean. Asterisks represent the significant difference between the samples and the untreated control.

**Figure 4 jcm-11-06634-f004:**
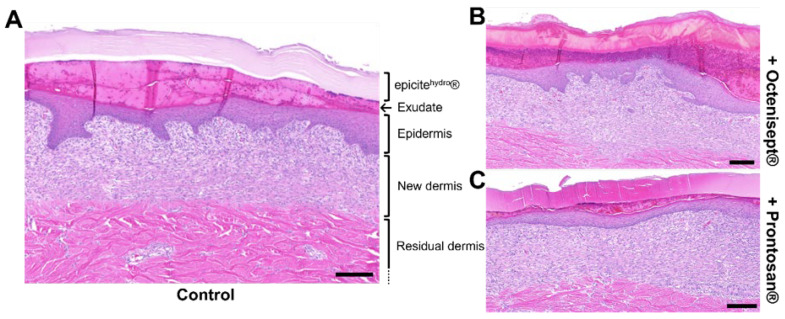
Histological aspects of the regenerated tissue. H&E stainings of representative examples of the regenerated tissue after 7 days of treatment with epicite^hydro®^ (n = 16 wounds) (**A**), epicite^hydro®^ combined with Octenisept^®^ (n = 24 wounds) (**B**) and combined with Prontosan^®^ (n = 24 wounds) (**C**). Scale bars representing 200 µm.

**Figure 5 jcm-11-06634-f005:**
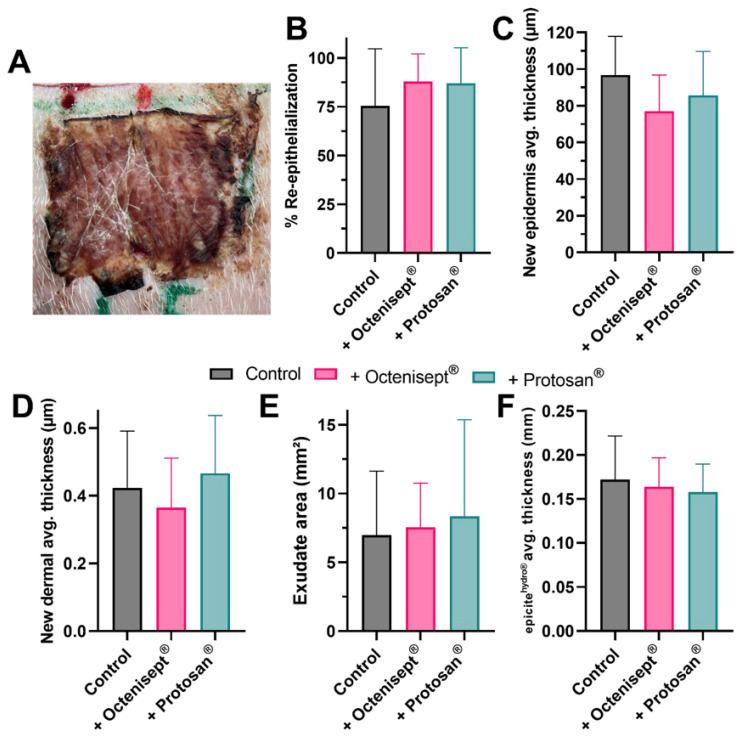
Influence of a 7-day treatment using antiseptics on the wound healing process outcome. Representative image of an epicite^hydro®^ dressing applied over the wound bed after the 7-day treatment. (**A**). Bar diagrams comprising the results for the percentage of re-epithelialization (**B**), average new epidermal thickness (**C**), average new dermal thickness (**D**), exudate area at day 7 (**E**) and epicite^hydro®^ dressing average thickness at day 7 (**F**) for the wounds treated using epicite^hydro®^ alone or combined with Octenisept^®^ or with Prontosan^®^. The results are presented as mean and standard deviation of the mean.

## Data Availability

All data generated or analyzed during this study are included in this published article.

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
