# Peer review of "The Impact of Antiseptic-Loaded Bacterial Nanocellulose on Different Biofilms—An Effective Treatment for Chronic Wounds?"

_jcm, 2022, doi:10.3390/jcm11226634_

Round 1

Reviewer 1 Report

The work is interesting and well done but the introduction, discussion and references need to be implemented. It would be interesting to evaluate the release of other antibacterial substances in the future.

Indroduction

What are the stages of tissue repair?

Please better explain the concept of infection, colonisation  and critical colonisation.

Debridement is an important step for reducing biofilm and bacterial load.

What are the techniques to assess biofilm?

What are the standard therapies to treat infection and biofilm?

Biofilm is typical of a chronic wounds.

Discussion

Explain better the standard wound treatments. Treatments depend on aetiology and wound characteristics.

The choice of dressing depends on the wound and perilesional skin features. The acronym TIME and the principles of wound bed preparation are used for wound assessment and secondary management.

hyperbaric oxygen therapy, low-level laser therapy and electrical stimulation are not standard therapies but used in specific cases.

Negative pressure therapy is indicated for deep chronic and very exudating wounds.

For biologics do you mean biologic drugs? Biologic drugs have indication mainly in inflammatory wounds ( for example pyoderma gangrenosum )

Antibacterial dressings include iodine and biding bacteria dressings and more. Can you add the literature data in the discussion?

Author Response

  • The reviewer asked for an implementation of the introduction, discussion and references, while pointing out the interest of a future evaluation of the release of other antibacterial substances. We would like to thank you very much for the feedback and will add more information to these sections as mentioned in the following paragraphs. Regarding the release of other substances, we decided for this work to focus solely on two solutions whose uptake and release in vitro was already published by our group. Future projects aim to test other substances and we will report these results in future publications.

  • The reviewer stated, that an explanation about the stages of tissue repair is missing in the introduction section. We added the following paragraphs to give important key information:

Page 3:

The biological process of successful wound healing is achieved through four precisely programmed, consecutive phases: hemostasis, inflammation, proliferation, and remodeling [1].

  • According to the reviewers comment regarding the concept of infection and colonization, as well as the presence of biofilm in chronic wounds we adapted and added to the paragraphs as follows:

Page 3:

While wounds may be colonized with a variety of microorganisms, tissue invasion or damage does not happen necessarily [3]

Shifts in the colonization flora however may cause pathogenic biofilms, which may be considered one of the most important factors contributing to pathological wound healing in addition to numerous potential factors like the patient's age, nutritional status, presence of a chronic disease or immunocompromised state [1], [2]. The presence of pathogenic biofilm may also be accompanied with infection, causing local symptoms such as swelling, erythema, pain or heat [3]

  • The reviewer highlighted the role of debridement in the reduction of bacterial load and standard therapies.

Thank you very much for pointing out this important fact – we have changed the paragraph concerned as follows:

Page 3:

While surgical debridement is an effective option in the reduction and eradication of bacterial load [7], non-surgical attempts to eradicate pathogenic biofilms and treat chronic wound infections are limited to the use of conventional antibiotics and antiseptics to date [5].

  • According to the missing statement about techniques to assess biofilm, the following passage has been added:

Page 3:

Characterization of this biofilm includes a variety of techniques ranging from older established methods (e.g. wound swabs, counting of bacterial colonies) to modern technologies such as fluorescent labeling and mathematical predictive modeling [4]

  • Thank you for your feedback on standard wound treatments, which were discussed more detailed within the discussion section.

Page 12:

The current standard of care for chronic wounds includes wounds swabs, cleaning, dressing and, if  necessary, debridement of the wound bed [14].

In addition, environment sensors , using pH, hydration, odor or optical sensors are available to monitor and manage biofilms and detect early changes in the wound bed associated with pathological wound healing [14]. New strategies of therapeutic intervention may also include the targeting of the wound microenvironment [14]. However,  targeting and eradicating pathogenic biofilms is challenging, since biofilms can be highly tolerant and resistant to antibiotics and antiseptics [2], [15]. 

  • The reviewer stated, that the choice of dressing depends on the wound and perilesional skin features and highlighted the acronym TIME and the principles of wound bed preparation in this context.

Since the wound bed assessment and its preparation is crucial to enhance the wound healing outcome, the following paragraphs were added:

Page 12:

As another, well established approach, the dynamic concept of wound bed preparation is particularly beneficial in chronic wounds, that fail to progress [15] This concept comprises comprehensive strategies of Tissue management, Inflammation and infection control, Moisture balance and Epithelial advancement (T.I.M.E) to maximize the wound healing potential [15].

However,  especially infection control via targeting and eradicating pathogenic biofilms is challenging, since biofilms can be highly tolerant and resistant to antibiotics and antiseptics [2], [16].

  • The concern was raised, that hyperbaric oxygen therapy, low-level laser therapy and electrical stimulation are not standard therapies but used in specific cases, and negative pressure therapy is indicated for deep chronic and very exudating wounds. We changed these statements as follows, furthermore, “biologics” was a wrong description which was also corrected:

Page 17:

In addition to delivery of antibiotics, physical treatment approaches such as hyperbaric oxygen therapy, low-level laser therapy and electrical stimulation can be used in specific cases to accelerate the healing of chronic ulcers, although inconsistent success is reported [42].

Negative pressure wound therapy is a different physical treatment approach indicated for deep chronic and very exudating wounds, based on differential suction or vacuum creation onto the wound bed to enhance fluid removal, reduce edema and alter the wound microenvironment for subsequently skin grafting [43].

  • Thank you very much for pointing out, that antibacterial dressings include different forms – according to your comment, the following references were added:

  1. de Mattos IB, Holzer JCJ, Tuca A-C, Groeber-Becker F, Funk M, Popp D, et al. Uptake of PHMB in a bacterial nanocellulose-based wound dressing: A feasible clinical procedure. Burns : journal of the International Society for Burn Injuries. 2019;45(4):898-904.

  1. Bernardelli de Mattos I, Nischwitz SP, Tuca A-C, Groeber-Becker F, Funk M, Birngruber T, et al. Delivery of antiseptic solutions by a bacterial cellulose wound dressing: Uptake, release and antibacterial efficacy of octenidine and povidone-iodine. Burns : journal of the International Society for Burn Injuries. 2020;46(4):918-27.

Reviewer 2 Report

The manuscript titled “The Biofilm Diminishing Effect of Bacterial Nanocellulose – an Effective Treatment for Chronic Wounds?” is a well-designed work with interesting results and possible future applications. I have some major and minor concerns that need to be answered by the authors:

Majors concern:

In the in vitro assays, why don't the authors used the commercial presentation of both antiseptics (for example as gel presentation) for a better comparison of the results obtained? Does BNC with antiseptic work better than antiseptic alone?     

In the same way, in the in vivo assays, why do not the authors just use the antiseptic (without BNC) as controls? Or untreated control for a better understanding of performance.

Minor concern:

In Materials and Methods, the phrases "The efficacy of the biofilm" (indicated in the text) should be changed.

In 2.5 (Test in vivo) I do not understand the number of wounds per animal. The authors reported 24 wounds in total, 12 with each antiseptic but a further 8 wounds as a control using only epicite. It must be explained.

Discussion: This title is out of place.

During the text, in some parts, there are different font sizes and some grammatical errors (some indicated in the manuscript).

Author Response

  • The reviewer asked, why no commercial presentation of both antiseptics (e.g. as gel presentation) for a better comparison of the results obtained were used in in-vitro

Thank you for your feedback on these points. In previous works (1, 2), we were able to show a sustained delivery of the antiseptic solutions in vitro, a feature that the solution alone cannot offer. Additionally, in our in-vitro tests against Staphylococcus aureus, the loaded BNC achieved superior results in comparison to our positive controls, including the octenidine antiseptic in gel preparation. Those were the main reasons for us to decide for the presented approach. Additionally, during our experimental design, we decided to focus on a comparison between the loaded BNC and another dressing ready to treat the infection. In this case, we compared to silver-containing dressings, which is known to offer good results against microorganisms capable of producing biofilms. The aforementioned information was added to the methods section.

  1. 1. de Mattos IB, Holzer JCJ, Tuca A-C, Groeber-Becker F, Funk M, Popp D, et al. Uptake of PHMB in a bacterial nanocellulose-based wound dressing: A feasible clinical procedure. Burns : journal of the International Society for Burn Injuries. 2019;45(4):898-904.

  1. Bernardelli de Mattos I, Nischwitz SP, Tuca A-C, Groeber-Becker F, Funk M, Birngruber T, et al. Delivery of antiseptic solutions by a bacterial cellulose wound dressing: Uptake, release and antibacterial efficacy of octenidine and povidone-iodine. Burns : journal of the International Society for Burn Injuries. 2020;46(4):918-27.

  • Concerns were also raised regarding the control groups. In our in vivo assays, we decided to establish a comparison focused solely on the effect of the prolonged exposure of antiseptic solutions to the wound bed. This is a feature offered by the combination of the BNC and the antiseptics. The antiseptics alone would not be able to maintain a sustained release into the wounded tissue. Additionally, the decision to use the BNC in all treatments, was to maintain our focus on the influence of the antiseptics. To analyze whether the solutions would impact any of the measured wound healing parameters. An untreated control would add an analysis of the wound healing promoted by the BNC itself.

  • According to the reviewers comment, the following statement was adapted:

Page 5:

The effect of loaded BNC on the biofilm was tested against a 24-hour biofilm of Gram-positive bacteria (Staphylococcus aureus) as well as fungi (Candida albicans) and a 48-hour biofilm of Gram-negative bacteria (Pseudomonas aeruginosa). A biofilm assay that was left untreated was used as negative control.

  • The reviewer indicated a calculation error in section 2.5, since 24 wounds were reported. Thank you very much, indeed, we miscalculate the total amount of wounds. It was actually a total of 32 wounds: 8 wounds treated with the control and 12 treated with each antiseptic. This error was corrected as follows:

Page 8:

A total of 32 wounds, divided in the two animals, were treated using this approach. 12 wounds were treated using epicitehydro incubated with Octenisept® and 12 treated with the combination with Prontosan®. As control, 8 wounds, 4 in each animal, were treated solely using the using epicitehydro dressings.

  • The reviewer noted, that the title is inappropriate. Therefore, we changed it to the following: “The Impact of Antiseptic-loaded Bacterial Nanocellulose on Different Biofilms – an Effective Treatment for Chronic Wounds? “

  • Several different font sizes and grammatical errors were found, which were corrected and font size was standardized.

Round 2

Reviewer 2 Report

I consider that the manuscript is ready for publication